# A Review of Recent Advances in Chromatographic Quantification Methods for Cyanogenic Glycosides

**DOI:** 10.3390/molecules29204801

**Published:** 2024-10-11

**Authors:** Yao Zhao, Shuai Wen, Yan Wang, Wenshuo Zhang, Xiangming Xu, Yi Mou

**Affiliations:** College of Pharmacy and Chemistry & Chemical Engineering, Taizhou University, Taizhou 225300, China; 15996256893@163.com (Y.Z.); wen15298503462@126.com (S.W.); wangyan213072802@163.com (Y.W.); wszhang2023@163.com (W.Z.);

**Keywords:** cyanogenic glycosides, chromatography, quantification

## Abstract

Cyanogenic glycosides are naturally occurring compounds found in numerous plant species, which can release toxic hydrogen cyanide upon hydrolysis. The quantification of cyanogenic glycosides is essential for assessing their potential toxicity and health risks associated with their consumption. Liquid chromatographic techniques coupled with various detectors have been widely used for the quantification of cyanogenic glycosides. In this review, we discuss recent advances in chromatographic quantification methods for cyanogenic glycosides, including the development of new stationary phases, innovative sample preparation methods, and the use of mass spectrometry. We also highlight the combination of chromatographic separation with mass spectrometric detection for the identification and quantification of specific cyanogenic glycosides and their metabolites in complex sample matrices. Lastly, we discuss the current challenges and future perspectives in the development of reliable reference standards, optimization of sample preparation methods, and establishment of robust quality control procedures. This review aims to provide an overview of recent advances in chromatographic quantification methods for cyanogenic glycosides and their applications in various matrices, including food products, biological fluids, and environmental samples.

## 1. Introduction

Cyanogenic glycosides are a group of naturally occurring compounds found in various plants, particularly in seeds and leaves [1]. These compounds consist of a sugar molecule (glycoside) attached to a cyanide group (Figure 1). When these glycosides are hydrolyzed, they release hydrogen cyanide (HCN), a highly toxic substance [2,3].

Cyanogenic glycosides serve as a defense mechanism for plants against herbivores and pathogens [4,5]. The release of HCN upon tissue damage acts as a deterrent to protect the plant. Common sources of cyanogenic glycosides include cassava, almonds, apricot kernels, cherry laurel, and bamboo shoots [1,6]. The levels of cyanogenic glycosides can vary among different plant species and even within different parts of the same plant. However, some animals have evolved mechanisms to detoxify or tolerate cyanogenic glycosides, enabling them to consume these plants without harm [5,7].

Consuming foods that are high in cyanogenic glycosides can be dangerous if not properly processed or cooked. It is important to note that cooking, soaking, or fermenting these foods can reduce their cyanide content to safe levels [8,9,10]. However, improper preparation or consumption of large quantities of these compounds can lead to cyanide poisoning, which can cause symptoms such as headache, dizziness, nausea, and in severe cases, it can be lethal [6,11,12,13]. Therefore, it is crucial to be aware of the presence of cyanogenic glycosides in certain foods and to follow proper preparation methods to ensure their safe consumption.

During the past 15 years, there have been significant advancements in the development of quantification methods for cyanogenic glycosides. These methods aim to accurately measure the levels of cyanogenic glycosides in various samples, including foods, plants, and biological fluids. Overall, the development of quantification methods for cyanogenic glycosides has focused on improving their sensitivity, selectivity, and accuracy. These advancements allow for better understanding, monitoring, and control of cyanogenic glycoside levels in various samples, contributing to food safety, plant breeding, and toxicology research.

The aims of this review are to (a) detail the existing chromatographic methods available, focusing on cyanogenic glycosides analysis since 2008, and (b) discuss potential directions for chromatographic and non-chromatographic technologies to further advance methods in terms of analysis time, sensitivity, specificity, and target analytes for the analysis of cyanogenic glycosides. Around 40 chromatographic and non-chromatographic methods for quantification of cyanogenic glycosides are covered in this review, with a strong emphasis on method development and a user-friendly approach.

## 2. Toxicity of Cyanogenic Glycosides in Plants

Many different cyanogenic glycosides have been identified in different plant species, each with their own specific characteristics, distribution, and potential health implications. In this review, we will focus on widely distributed and well-studied cyanogenic glycosides for their toxicity and determination methods, including amygdalin, linamarin, prunasin, and dhurrin.

### 2.1. Amygdalin

Amygdalin is a cyanogenic glycoside found in the seeds of various fruits, including apricots, peaches, and bitter almonds [14,15]. β-glucosidase often plays a role during the degradation of amygdalin. This enzymatic reaction leads to the breakdown of amygdalin into glucose, benzaldehyde, and HCN [16].

HCN is a highly toxic compound and acts as a potent inhibitor of cellular respiration. It binds to cytochrome oxidase in the mitochondria, preventing the utilization of oxygen in the electron transport chain [17,18,19]. This disruption of cellular respiration leads to cellular hypoxia and can have severe toxic effects on various organs and tissues in the body.

### 2.2. Linamarin

Linamarin is a cyanogenic glycoside that is commonly found in plants such as cassava (Manihot esculenta) and lima beans (Phaseolus lunatus) [20,21]. It is structurally similar to amygdalin and is known for its potential toxicity when ingested. Linamarin breaks into glucose and acetone cyanohydrin under the catalysis of β-glucosidas [22]. Acetone cyanohydrin is a cyanide precursor that can further release HCN under certain conditions [22]. The toxic effects of linamarin are primarily attributed to the toxicity of the HCN released upon its breakdown.

### 2.3. Prunasin

Prunasin is a cyanogenic glycoside found in various plant species, including cherry (*Prunus* spp.) and almond (*Prunus dulcis*) seeds [23,24]. Similar to other cyanogenic glycosides, prunasin can release mandelonitrile when hydrolyzed, which can further release HCN under certain conditions [25]. The specific toxicity of prunasin itself and its direct administration in humans or animals is not extensively studied. Most research focuses on the toxicity of cyanide released from prunasin rather than the direct administration of prunasin itself.

### 2.4. Dhurrin

Dhurrin is a cyanogenic glycoside found in various plant species, including sorghum (Sorghum bicolor) and some bamboo species [26,27]. It is similar to other cyanogenic glycosides in that it can release HCN, with p-hydroxyphenylacetaldehyde cyanohydrin serving as the intermediate [28].

## 3. Determination of Cyanogenic Glycosides Using HPLC/UPLC

Various high performance liquid chromatography (HPLC) methods have been developed for the quantification of amygdalin, prunasin, linamarin, and other related compounds in fresh plants and processed plant products for over two decades, with the ability to separate, identify, and quantitate each component in a mixture. Table 1 shows a list of existing HPLC methods (except those coupled to mass spectrometry), focusing on those reported in last 15 years for analyzing cyanogenic glycosides in fresh and processed plant samples. Sample processing methods, columns, detectors, and limits of detection (LODs) are described in this section.

Prior to HPLC analysis, an extraction procedure needs to be performed to isolate and dissolve the target analytes. In most instances, cyanogenic glycosides were extracted from fresh and processed plant samples with an organic phase and water in different volume ratios by sonicating, shaking, and refluxing. Guillermo Arrázola [29] treated almond seed extracts with polyvinylpolypirrolidone or activated carbon for removal of pigments. Wang et al. [30] used macroporous adsorption resins for adsorption and separation of amygdalin from thinned bayberry kernels crude extracts. Meanwhile, hydrolysis of cyanogenic glycosides was used during sample pretreatment, especially when water was used for the extraction of cyanogenic glycosides. For example, emulsin, an enzyme hydrolyzing amygdalin, is present in Armeniacae Semen, leading to the epimerization of amygdalin during sample processing. Kwon et al. [31] diminished the effect of emulsin by increasing the heating temperature and reducing the heating time during the extraction procedure for better accuracy.

As can be seen in Table 1, reverse-phase HPLC columns were mainly utilized for the determination of cyanogenic glycosides. When operating on C18 columns, mobile phases were often acidified to achieve better retention and separation of cyanogenic glycosides. For example, eluents containing formic acid [32], trifluoroacetic acid [33], phosphoric acid [34], and phosphate buffer at low pH [31,35] were used to achieve separation. A UPLC C18 column with a smaller particle size and higher pressure was also used to achieve a better resolution and faster analysis in the analysis of cyanogenic glycosides [33]. Furthermore, a hydrophilic interaction liquid chromatography (HILIC) column (Ascentis Express OH5 column) can be employed in the quantification of cyanogenic glycosides with faster separation (retention time of amygdalin at 2.60 min) considering the sugar structure in glycosides [36].

Quantification of cyanogenic glycosides was mainly achieved by monitoring the absorbance at an ultraviolet (UV) range using a diode array detector (DAD) or UV detector (UVD) (Table 1). In most circumstances, detections were performed at 210 nm to 220 nm, with LODs for cyanogenic glycosides at sub-µg/mL. Xu et al. [35] exceptionally monitored amygdalin at 254 nm, leading to a relatively low LOD for amygdalin at 2 µg/mL. Other than quantification, the UV spectrum played a role in the structural characterization as well. For example, Havlíková, L., et al. [36] identified amygdalin in extracts by comparing the retention times and UV spectra with standard amygdalin.

Recent HPLC methods were modified for broader application and lower detection limits with the combination of multifarious sample preparation methods. Wang et al. [37] purified mouse plasma with solid-phase extraction (SPE) prior to HPLC separation, achieving LODs of 0.02 μg/mL for amygdalin and 0.03 μg/mL for prunasin and LOQs of 0.04 μg/mL for amygdalin and prunasin in rat plasma.

## 4. Determination of Cyanogenic Glycosides Using LC-MS

Although mass spectrometry (MS) is fairly expensive to carry out, exploration with MS is thriving due to its extremely high sensitivity and selectivity. LC-MS methods were applied for both qualification and quantification studies of cyanogenic glycosides in fresh and processed plant samples and biological samples (Table 2). In this section, MS methods operating on different modes are classified and discussed.

MS methods have been utilized for structural characterization of cyanogenic glycosides operating on MS/MS mode with the advantages of increased specificity, lower detection limits, fragmentation patterns, and isomer differentiation compared to single-stage MS alone. Xu et al. [32] operated HPLC-ESI-MS/MS analyses in negative mode and identified amygdalin, neoamygdalin, and amygdalin amide by comparing the retention times and mass data in different processed bitter almonds. Ivan M. Savic et al. [40] confirmed the isolated amygdalin from plum seeds (Pruni domesticae semen) by comparing the obtained MS/MS spectrum with existing publications.

Quantification analyses of cyanogenic glycosides were usually obtained using multiple reaction monitoring (MRM) mode, with some exceptions. As shown in Table 2, the LODs of the LC-MS methods developed for quantification of cyanogenic glycosides during the 15 years can reach the level of ng/mL. Appenteng et al. [42] carried out the simultaneous quantification of amygdalin, dhurrin, prunasin, and linamarin with MRM. Li et al. [43] performed negative-mode ESI using MRM, with mass transitions at *m*/*z* 457.2–279.1, for detecting amygdalin in rat plasma. Li et al. [44] utilized the positive MRM of the transitions of *m*/*z* 475.2→163.1 for amygdalin and *m*/*z* 498.2→179.1 for paeoniflorin. Wang et al. [45] analyzed linustatin and neolinustatin in flaxseed powder with negative-mode ESI, and the transitions were set at 408–323 and 422–323, respectively. Mateja Senica et al. [46] targeted amygdalin and prunasin in apricot and cherry fruit kernels and liqueur by operating in positive MRM mode with mass transitions of 480→347, 328 for amygdalin and 318→128, 185 for prunasin.

Apart from the quantification analyses of cyanogenic glycosides achieved using MRM mode, there were other quantification methods using LC-MS. Jandirk Sendker et al. [47] quantitated prunasin and related compounds in the leaves of Prunus laurocerasus using a time-of-flight mass spectrometer equipped with an Apollo electrospray ionization source in positive mode. Nanna Bjarnholt et al. [48] determined linamarin in Parnassius (Papilionidae) butterflies and their food plants by MS analyses.

LC-MS methods of cyanogenic glycosides can be applied to biological samples owing to their extremely high sensitivity. As shown in Table 2, LC-MS methods for detecting cyanogenic glycosides were developed for analysis in rat and human plasma. During sample pretreatment, deproteinization of plasma was achieved by SPE or protein precipitation. Li et al. [43] monitored the amygdalin concentration in rat plasma after oral dosage, predicting the half-life period of amygdalin in rat plasma. Li et al. [44] developed an LC-MS/MS method for detecting amygdalin in human plasma and applied it to a pilot pharmacokinetic study of amygdalin in healthy volunteers after intravenous infusion administration of Huoxue-Tongluo lyophilized powder for injection. Song et al. [50] reported that stereo-selective metabolism of amygdalin occurred in vivo by monitoring the epimers of amygdalin and prunasin in rat plasma.

## 5. Determination of Cyanogenic Glycosides without HPLC Separation

Beyond the quantification of cyanogenic glycosides, numerous methodologies exist for the measurement of HCN, which is the toxic agent released by these glycosides. Despite our primary focus on chromatographic techniques for the analysis of cyanogenic glycosides, it is pertinent to acknowledge that alternative methods are employed for the quantification of HCN. Consequently, we will also include these alternative approaches within the scope of our review, recognizing their importance and applicability in assessing HCN levels.

Several assays developed for the determination of HCN derived from cyanogenic glycosides are listed in Table 3. During the assays, HCN was liberated from cyanogenic glycosides through hydrolysis, trapped in appropriate solutions, and reacted with sensors. Although they have limited selectivity, these assays demonstrate their superiority for a laboratory with limited equipment and for field analysis.

An infrared (IR) spectrometer can be employed for the identification and quantification of cyanogenic glycosides. Ivan M. Savic et al. [40] confirmed the structure of separated amygdalin from plum seeds based on the IR spectroscopic method, with the presence of the functional groups of amygdalin being confirmed in the isolate. Victoria Cortés et al. [56] used near-infrared (NIR) spectroscopy to predict the amygdalin content in intact almonds by applying partial least squares to the spectral data.

Moreover, a Raman microscope can be applied in the detection of cyanogenic glycosides. Philip Heraud et al. [57] located cyanogenic glucosides in the cytoplasm in sorghum cells through Raman microspectroscopic mapping. The analysis was conducted in intact tissues to avoid possible perturbations and imprecision that may accompany methods that rely on bulk tissue extraction methods, such as protoplast isolation.

## 6. Future Perspectives

Despite the significant progress in the development of chromatographic methods for the quantification of cyanogenic glycosides, there are still challenges that need to be addressed, including the development of reliable reference standards, the establishment of robust quality control procedures, and unified safety standards for cyanogenic glycosides in food.

Cyanogenic glycosides are a diverse group of compounds with various structures, which can make the synthesis or purification of a specific reference standard difficult and costly. Junko Tsukioka et al. [54] synthesized a taxiphyllin standard following previous methods, and Ryota Akatsuka et al. [55] extracted prunasin standard from plants. However, it can be very difficult to obtain cyanogenic glycosides standards which may be rarely studied, at a low concentration, or with poor stability, hindering accurate quantification of cyanogenic glycosides.

In light of the current case-by-case evaluation approach employed by the U.S. Food and Drug Administration (FDA) for monitoring the safety of amygdalin in stone fruit kernel products, it is imperative that we underscore the need for the establishment of robust quality control procedures and unified safety standards for cyanogenic glycosides across the food industry. Such standards are essential for ensuring consistent and reliable quantification of these compounds, which is critical for protecting public health. By advocating for the adoption of these standardized chromatographic techniques, we can achieve consistent and reliable quantification of cyanogenic glycosides, thereby facilitating the creation of a cohesive and stringent set of safety standards that will protect consumers and promote public health.

## 7. Conclusions

In conclusion, the quantification of cyanogenic glycosides has gained significant attention due to their potential toxicity and the health risks associated with their consumption. In this review, recent advances in chromatographic quantification methods for cyanogenic glycosides have been discussed, mainly focusing on HPLC-UV and HPLC-MS. The development of new stationary phases, innovative sample preparation methods, and the use of mass spectrometry have contributed to the improvement of the sensitivity, selectivity, and accuracy of these methods.

## Figures and Tables

**Figure 1 molecules-29-04801-f001:**
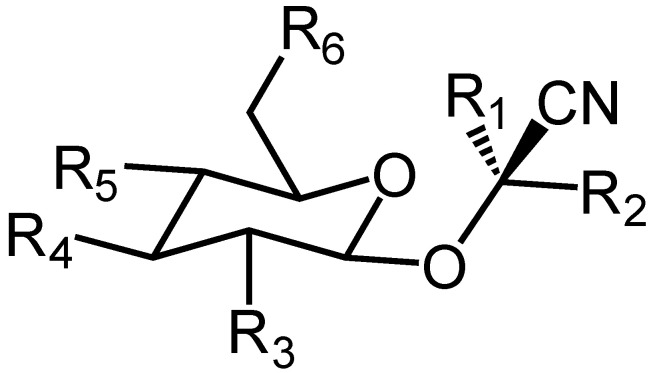
General structure of cyanogenic glycosides. In the structure formula, R_1_ represents a proton for amygdalin, prunasin, and dhurrin and a methyl group for linamarin, while R_2_ is a changeable organic group.

**Table 1 molecules-29-04801-t001:** Comparison of HPLC/UPLC methods for the determination of cyanogenic glycosides.

Analytes	Matrices	Columns	Eluent	Flow Rate (mL/mL)	Retention Time (min)	Detector	Absorption Wavelength (nm)	LOD	Ref.
Amygdalin	Almond seeds	Waters Symmetry column 250 mm × 4.6 mm	Acetonitrile and water (20:80, *v*:*v*)	1.5	-	UVD	218	-	[29]
Amygdalin	Thinned bayberry kernels	Symmetry^®^C18 MG column (4.6 mm × 250 mm, 5 µm)	Acetonitrile and water (13:87, *v*:*v*)	1	-	DAD	214	-	[30]
D-Amygdalin	Armeniacae Semen powder	Luna 3u C18 (2) 100A (4.6 mm × 150 mm, 3 μm, Phenomenex)	10 mM sodium phosphate buffer (pH 2.3) containing 13.5% acetonitrile	0.6	~12	UVD	214	5 μM per amount injected	[31]
		Capcell Pak C18 MG (4.6 mm × 250 mm, 5 μm, Shiseido)	10 mM sodium phosphate buffer (pH 3.1) containing 8.5% acetonitrile	1.2	~31				
Amygdalin amide, D-amygdalin, and L-amygdalin	Processed bitter almonds	Eclipse EXTEND C18 column	Methanol and 0.1% formic acid in water	1	5.4, 11.5 and 16.0	DAD	254	2 µg/mL	[32]
Prunasin and amygdalin	Saskatoon berries	Zorbax SB C18 150 mm × 2.1 mm with a 12.5 mm × 2.1 mm guard column of the same material	2 mM trifluoroacetic acid and methanol	0.3	5.13 and 6.64	DAD	210	0.5 mg/kg and 1.5 mg/kg FW of berries	[33]
Amygdalin	Organs and tissues of loquat fruit	ODS C18 column (250 mm × 4.6 mm, 5 μm)	Methanol and 0.03 mol/L phosphate buffer (pH 2.8) (25:75, *v*:*v*)	1	-	UVD	210	-	[34]
Amygdalin	Apple seeds, fresh apples, and processed apple juices	Phenomenex C18, Type Nucleosile 3, 120 A (150 mm × 4.60 mm, 3 µm)	Methanol and water (25:75, *v*:*v*)	1	~4	DAD	214	0.1 µg/mL	[35]
Amygdalin	Food supplements	Ascentis Express OH5 column (100 mm × 3.0 mm; 2.7 μm)	Acetonitrile and 10 mM ammonium acetate pH 3.8 (90:10, *v*:*v*)	0.8	2.6	UVD	215	0.150 mg/L	[36]
Amygdalin and prunasin	Mouse plasma	C18 5μ ODS column	0.05% formic acid in acetonitrile, *v*/*v*; and 0.05% formic acid in water, *v*/*v*	1	3.14 and 6.82	UVD	215	-	[37]
Amygdalin	Fruit kernels, seeds, and processed products	Phenomenex C18, Type Nucleosile 3, 120 A (150 mm × 4.60 mm, 3 µm)	Methanol and water (25:75, *v*:*v*)	1	~4	DAD	214	0.1 µg/mL	[38]
Linamarin, linustatin, and neolinustatin	Flaxseed meal from twenty-one varieties	Kinetex 4.6 mm × 125 mm, 2.6 micron C-18	0.05% Phosphoric acid in HPLC grade water and methanol	-	4.9, 8.9 and 18.8	UVD	230	-	[39]
Amygdalin	Plum seeds	SUPELCO Analytical HS-C18 column	Acetonitrile and water (75:25, *v*:*v*)	0.9	-	UVD	215	-	[40]
Prunasin	Roots of Prunus serotina and Quercus petraea	SunFireTM C18 column (4.6 mm × 250 mm, 5 μm) together with the precolumn Waters (4.6 mm × 20 mm)	Methanol and water (15:85, *v*:*v*)	1.5	-	UVD	218	-	[41]

**Table 2 molecules-29-04801-t002:** Comparison of HPLC/UPLC-MS methods for the determination of cyanogenic glycosides.

Analytes	Matrices	Sample Processing	Columns	Eluent	Flow Rate (mL/mL)	Retention Time (min)	Detector	LOD	Ref.
Amygdalin amide, D-amygdalin, and L-amygdalin	Processed bitter almonds	Extraction with 70% aqueous methanol	Eclipse EXTEND C18 column	Methanol and 0.1% formic acid in water	1	5.4, 11.5 and 16.0	ESI- using MS/MS	-	[32]
Amygdalin, dhurrin, prunasin, and linamarin	American elderberry	Extraction with 75% methanol	C18 column (Acquity BEH, 1.7 μM, 2.1 mm × 50 mm, Waters)	Water with 0.1% formic acid and acetonitrile with 0.1% formic acid	0.2	4.61, 2.54, 5.37, and 1.18	ESI+ using SIR	3, 3, 3, and 1 ng/mL	[42]
Amygdalin	Rat plasma	SPE	Gemini C18 analytical column (50 mm × 2.0 mm, 5 µm; Phenomenex)	Methanol and water (85:15; *v*/*v*)	0.25	1.68	ESI- using MRM	1.25 ng/mL	[43]
Amygdalin and paeoniflorin	Human plasma	Protein precipitation with methanol	Hedera ODS-2 analytical column (2.1 mm × 150 mm, 5 µm; Hanbon Science and Technology)	Acetonitrile and 5 mM ammonium acetate buffer solution containing 0.05% formic acid (20:80, *v*/*v*)	0.3	2.14 and 3.18	-	-	[44]
Linustatin and neolinustatin	Flaxseed powder	Extraction with methanol and purification with a silica gel column	Agilent ZORBAX Eclipse Plus C18 (2.1 mm × 50 mm; particle size, 1.8 μm)	Water containing 0.1% formic acid (*v*/*v*) and acetonitrile	0.4	1.8 and 2.7	ESI- using MRM	-	[45]
Amygdalin and prunasin	Apricot kernel, apricot liqueur, cherry kernel, and cherry liqueur	Extraction with 70% methanol	HYPERSIL GOLD aQ column (Thermo Scientific)	Water with 0.1% formic acid with 3% methanol (*v*/*v*/*v*) and methanol with 0.1% formic acid with 3% water (*v*/*v*/*v*)	0.8	3.85 and 4.4	ESI+ using MRM	-	[46]
Prunasin and related compounds	Leaves of *Prunus laurocerasus*	Extraction with 70% acetonitrile	Dionex Acclaim RSLC 120 C18 column (2.1 mm × 100 mm, 2.2 μm)	Water with 0.1% formic acid and acetonitrile with 0.1% formic acid	0.8	-	ESI+	-	[47]
Linamarin	Parnassius butterflies and their food plants	Extraction with 55% methanol containing 0.1% formic acid	Zorbax SB-C18 column (Agilent; 1.8 μM, 2.1 mm × 50 mm)	Water with 0.1% (*v*/*v*) HCOOH and 50 mM NaCl and acetonitrile with 0.1% (*v*/*v*) HCOOH	0.2	2.6	ESI+	-	[48]
Prunasin and amygdalin	Almond and sweet cherry	Extraction with 85% methanol	Zorbax SB-C18 column (Agilent; 1.8 mm, 2.1 mm × 50 mm)	Water with 0.1% (*v*/*v*) HCOOH and 50 mM NaCl and acetonitrile with 0.1% (*v*/*v*) HCOOH	0.2	7 and 6.6	ESI+	-	[49]
Epimers of amygdalin and prunasin	Rat plasma	Protein precipitation with acetonitrile	Zorbax SB-C18 column (100 mm × 3.0 mm, 1.8 μm; Agilent Technologies)	Acetonitrile and 0.1% formic acid aqueous solution	0.4	-	ESI+ using MRM	-	[50]
Linamarin	Cassava flour and tapioca	Extraction with acetonitrile	CAPCELL PAK AQ column (2.1 mm × 250 mm, Shiseido)	Water containing 0.01% acetic acid and methanol (70:30; *v*/*v*)	0.2	5.5	ESI- using MRM	0.75 and 0.84 µg/g	[51]
Linamarin, dhurrin, amygdalin, prunasin, and sambunigrin	American elderberry	Extraction with 80% methanol	ACQUITY UPLC HSS T3 column (1.8 μM, 2.1 mm × 100 mm)	Water with 2 mM ammonium formate and ACN	0.5	3.34, 6.40, 9.89, 11.14, and 11.34	ESI+ using MRM	1.2, 1.2, 8.2, 2.0, and 0.9 ng/L	[52]
Amygdalin	Apricot Kernels and Almonds	Extraction with methanol	Phenomenex Kinetex XB-C18 (2.6 μM, 2.1 mm × 100 mm)	10 mM ammonium formate/0.1% formic acid/water and 10 mM ammonium formate/0.1% formic acid/methanol	0.3	3.3	ESI- using MRM	0.8 µg/g	[53]
taxiphyllin	Leaves of Hydrangea macrophylla var. thunbergii	Extraction with methanol	Cosmosil C18-PAQ column (5 μM, 4.6 mm × 250 mm)	acetonitrile and water	1	8.5 min	ESI+ using Q1 scan		[54]
prunasin	Perilla frutescens	Extraction with methanol	COSMOSIL 5C18 MS II column (250 × 4.6 mm)	Methanol and water (14:86)	0.4	~34 min	ESI-	0.00157 mg/mL (LOQ)	[55]

**Table 3 molecules-29-04801-t003:** Comparison of methods for determination of HCN derived from cyanogenic glycosides.

Reference Standard	Matrices	Sample Processing	Principle	Detector Condition	LOD	Ref.
Amygdalin	Plum seeds	Extraction with ethanol	IR absorbance	absorbance measured over 4000–400 cm^−1^		[40]
Amygdalin	Intact almonds		NIR absorbance	absorbance measured over 4000–400 cm^−1^		[56]
Dhurrin	Cytoplasm in sorghum cells		Raman microspectroscopic mapping			[57]
Cyanohydrin and HCN contents	Transgenic Acyanogenic Kenyan cassava genotypes	Ectraction with acid extraction medium	RNA interference approach based on the König reaction	absorbance measured at 600 nm		[58]
Total cyanogenic glycosides	Fresh and processed cassava products	Extraction with 0.1 M phosphoric acid	Aquacyanocobyrinic acid chemosensor	absorbance measured at 578 nm		[59]
HCN	Commercial food products prepared from cassava	Homogenization with 0.1 M (25%, *v*/*v*, ethanol) phosphoric acid	IR absorbance	absorbance measured at 605 nm		[60]
HCN	Plants with resorcinol and picrate	Hydrolysis with 2 M sulfuric acid	IR absorbance	absorbance measured at 488 nm	0.05 µg/mL cyanide	[61]
Amygdalin	Persicae semen, Armeniacae semen, and Mume fructus	Sonication with methanol	1H-NMR spectrometry	Concentration corrected with certified reference material-grade bisphenol A		[62]
Total cyanide	Edible plants from 14 genera	Homogenization with 0.1 M phosphoric acid	Electrode cell Silver working electrode	0.00 V vs. Ag/AgCl reference	0.005 mg CN^−^/L.	[63]

## Data Availability

No new data were created.

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
