# Peer review of "A Review of Recent Advances in Chromatographic Quantification Methods for Cyanogenic Glycosides"

_molecules, 2024, doi:10.3390/molecules29204801_

Round 1

Reviewer 1 Report

Comments and Suggestions for Authors

The article, "A Review of Recent Advances in Chromatographic Quantitation Methods for Cyanogenic Glycosides" reviewed the chromatographic, as well as non-chromatographic methods of analysis of cyanogenic glycosides in various matrices such as foodstuffs and blood. This is an important review in the sense that cyanogenic glycosides are a potential threat to humans; and in ensuring food safety, we need to have sensitive analytical methods in place. This review article does that. However, I feel that improvements in the review article are needed in order to better its quality. I detail my comments below:

1. Can you summarise in the Introduction how many papers were reviewed, and the breakdown (e.g., how many researches used LC-DAD, LC-MS, etc)? Were there any previous reviews made on the chromatographic detection and quantitation of cyanogenic glycosides in plant materials and biological matrices? If so, what were their shortcomings that you wished to improve on? These information could clarify the impetus of your review.

2. The stated aims of the review were to: a) detail the existing chromatographic methods available from [the] perspective of cyanogenic glycosides analysis since 2008; and b) discuss potentials for chromatographic and non-chromatographic technologies.

a) Given Aim (a) I do not understand the need to discuss the determination of cyanogenic glycosides without HPLC separation, since this section did not discuss futures and potential of these technologies to the analysis of cyanogenic glycosides. In my opinion, this section can be confusing to the reader. I suggest that the authors focus on literature on chromatographic analyses of cyanogenic glycosides.

b) Since in the title, it mentions "Chromatographic Quantitation Methods", why did the authors choose to discuss also the futures of analysis of cyanogenic glycosides which includes that of non-chromatographic methods (e.g., IR, Raman, MALDI) (which is the point of discussion in Section 6)? As in Comment 2a, I suggest that the authors focus on the chromatographic methods and remove this section altogether.

3. In my opinion, the authors haven't fully maximised the literature on the chromatographic analysis of cyanogenic glycosides. A quick Scopus search using the terms "cyanogenic glycosides" and "chromatography" from 2008-present (including 2025) yielded around 119 articles (unfiltered), some of which were not included in the manuscript. I suggest adding these missing research to your review to enrich it further.

4. Sections 3 and 4, paragraph 1. I get that the authors wish to somehow introduce the section. However, the authors went straight to the recent developments (e.g., lowering of LOD in HPLC detection of cyanogenic glycosides in biological matrices) -- which I found rather confusing. I suggest that the authors outline instead what they wish to discuss throughout the section.

5. Section 3. I feel that paragraphs 3 and 4 just discussed general considerations in the HPLC analysis of cyanogenic glycosides, without pertaining to any research included in the review. A short discussion, a critical analysis, or an exposition of an exemplary paper would enrich this section further. Also, if there is a nice figure from the papers discussed in this section that could illustrate your point, that would be helpful. But if there is none, then no need to add a new figure.

6. Minor comment: I suggest that you incorporate the R1 and R2 of the 4 major cyanogenic glycosides in Figure 1. 

Comments on the Quality of English Language

In general, the quality of the English in the manuscript is good. I had no difficulty understanding the paper.

Reviewer 2 Report

Comments and Suggestions for Authors

After reviewing the manuscript titled " A Review of Recent Advances in Chromatographic Quantitation Methods for Cyanogenic Glycosides", I have the following comments:

1. The review focuses on an important subject: cyanogenic glycosides, which are toxic compounds found in plants, making the topic significant for food safety, toxicology, and analytical chemistry. The discussion of recent advances in chromatographic methods for quantifying cyanogenic glycosides is comprehensive and up-to-date, highlighting the evolution of these techniques, including HPLC, UPLC, and LC-MS methods.

2. There are several recent review papers on cyanogenic glycosides and their  quantification methods such as:

Tahir, Fizza, et al. "Cyanogenic glucosides in plant-based foods: Occurrence, detection methods, and detoxification strategies–A comprehensive review." Microchemical Journal (2024): 110065.

De Girolamo, Annalisa, Vincenzo Lippolis, and Michelangelo Pascale. "Overview of recent liquid chromatography mass spectrometry-based methods for natural toxins detection in food products." Toxins 14.5 (2022): 328.

Does the current manuscript provides any more new perspective than the ones already discussed in these review papers.

3. The manuscript could also delve deeper into the future challenges of cyanogenic glycoside analysis. While it touches on the need for better reference standards and sample preparation methods, it could expand on how these challenges are currently being addressed by the scientific community.

4. The paper could benefit from adding some references to very recent studies (from the last 1–2 years) to strengthen the timeliness of the review, particularly in the "Future Perspectives" section.

4. It may be beneficial to include a brief mention of the regulatory implications of cyanogenic glycoside quantitation, particularly in food safety contexts. For instance, which international bodies (like the FDA or EFSA) set safety limits for these compounds, and how the discussed methods can help ensure compliance with these standards.

5. Overall, this is a valuable and comprehensive review of recent advances in chromatographic methods for cyanogenic glycoside analysis. It provides a thorough discussion of the state-of-the-art techniques and highlights important areas for further investigation. With some refinement to improve clarity and add more critical discussion, this paper will serve as a strong resource for researchers in the field.

Comments on the Quality of English Language

1. "Quantitation" is a valid term in specific contexts, but "Quantification" is more commonly used. Ensure consistency with the rest of the document if "Quantitation" is intended.

2. In few instances, "essays" is used in place of "assays". This completely changes the meaning. 

3. Inconsistent capitalization in some reference entries (e.g., capitalizing each word in a title vs. using sentence case). Consistency is key in reference formatting.

Round 2

Reviewer 1 Report

Comments and Suggestions for Authors

I am happy with the changes made by the authors. Therefore I can recommend publication of the review to Molecules.

Comments on the Quality of English Language

English is still very much fine.